# Gait Variability at Different Walking Speeds

**DOI:** 10.3390/jfmk8040158

**Published:** 2023-11-08

**Authors:** Johnny Padulo, Susanna Rampichini, Marta Borrelli, Daniel Maria Buono, Christian Doria, Fabio Esposito

**Affiliations:** 1Department of Biomedical Sciences for Health (SCIBIS), Università degli Studi di Milano, 20133 Milan, Italy; susanna.rampichini@unimi.it (S.R.); marta.borrelli@unimi.it (M.B.); danielm.buono97@outlook.it (D.M.B.); christian.doria@unimi.it (C.D.); fabio.esposito@unimi.it (F.E.); 2IRCCS Galeazzi Orthopedic Institute, 20161 Milan, Italy

**Keywords:** human locomotion, symmetry, gait analysis, physiological response, kinematic analysis

## Abstract

Gait variability (GV) is a crucial measure of inconsistency of muscular activities or body segmental movements during repeated tasks. Hence, GV might serve as a relevant and sensitive measure to quantify adjustments of walking control. However, it has not been clarified whether GV is associated with walking speed, a clarification needed to exploit effective better bilateral coordination level. For this aim, fourteen male students (age 22.4 ± 2.7 years, body mass 74.9 ± 6.8 kg, and body height 1.78 ± 0.05 m) took part in this study. After three days of walking 1 km each day at a self-selected speed (SS) on asphalt with an Apple Watch S. 7 (Apple^TM^, Cupertino, CA, USA), the participants were randomly evaluated on a treadmill at three different walking speed intensities for 10 min at each one, SS − 20%/SS + 20%/ SS, with 5 min of passive recovery in-between. Heart rate (HR) was monitored and normalized as %HR_max_, while the rate of perceived exertion (RPE) (CR-10 scale) was asked after each trial. Kinematic analysis was performed, assessing the Contact Time (CT), Swing Time (ST), Stride Length (SL), Stride Cycle (SC), and Gait Variability as Phase Coordination Index (PCI). RPE and HR increased as the walking speed increased (*p* = 0.005 and *p* = 0.035, respectively). CT and SC decreased as the speed increased (*p* = 0.0001 and *p* = 0.013, respectively), while ST remained unchanged (*p* = 0.277). SL increased with higher walking speed (*p* = 0.0001). Conversely, PCI was 3.81 ± 0.88% (high variability) at 3.96 ± 0.47 km·h^−1^, 2.64 ± 0.75% (low variability) at SS (4.94 ± 0.58 km·h^−1^), and 3.36 ± 1.09% (high variability) at 5.94 ± 0.70 km·h^−1^ (*p* = 0.001). These results indicate that while the metabolic demand and kinematics variables change linearly with increasing speed, the most effective GV was observed at SS. Therefore, SS could be a new methodological approach to choose the individual walking speed, normalize the speed intensity, and avoid a gait pattern alteration.

## 1. Introduction

The analysis of human locomotion during activities such as pedaling [1,2], walking [3,4], race walking [5,6,7], and running [8,9,10,11,12] exhibits a repetitive and stereotypical movement pattern over time [13,14,15,16,17,18,19,20]. Numerous studies have focused on investigating the variability of the gait cycle paradigm (gait variability, GV) to better understand the bioenergetics and control of the human locomotion [21]. This seemingly simple activity [22,23,24,25,26,27] involves a complex task [28,29,30,31,32,33,34] requiring a precise synergy [35,36,37,38,39,40] between lower limb coordination [41,42,43,44,45,46,47] and muscle contraction [48,49,50,51,52,53,54] in response to both natural and non-natural conditions [55]. As a result, individuals need to continuously explore new strategies [56,57,58,59,60,61,62,63] and promptly adapt the motor task [64,65,66,67,68,69,70] to the immediate environment conditions, adjusting their footstep cycle to the most appropriate one.

In walking, stride-to-stride variability [71,72,73,74,75,76,77] arises due to the system’s constant need [78,79,80,81,82] to adjust inaccurate movements [83]. From a neurophysiological point of view, higher variability is associated with poor coordination level, while lower variability indicates better coordination level [84]. Pathological and non-pathological factors have been proven to affect the coordination level. Indeed, Parkinson’s disease, aging [55], and individuals with a lower limb injury [85] have been shown to exhibit high variability. Nonetheless, increased variability has been observed in healthy people due to changes in body positions during uphill running [86], alterations in body posture [1], and variations in walking speed [83]. Jordan et al. [83] showed that the better walking coordination level (i.e., lowest GV) in healthy young females occurred at walking speeds between 100 and 110% of the preferred walking speed. Even though the preferred speed favors better walking coordination [83], most of the studies on GV have used a lower speed to administer standardized speed on a treadmill [85,87,88]. Anyway, a divergence about the physiological effort between preferred walking speed on a treadmill compared to the overground [89] has been shown, suggesting that on a treadmill, the preferred walking speed was lower. Therefore, to obtain data that are more representative of daily activities, the GV should be studied at the preferred gait speed determined overground, as a treadmill does not fully represent the ground of daily activities. This methodological approach could mitigate the influence of the neurophysiological factors on bioenergetics variables, such as kinematic, kinetic, and motor control aspects. Contrarily, the motorized treadmill [90] provides the advantage of having long duration trials, such as those needed for gait variability study. Indeed, 400 steps are required for an accurate estimation of the step kinematic variability [91] at a constant speed. Certainly, treadmills offer researchers the advantage of precise control over walking speed, enabling extended trials with subjects confined to a limited motion capture space and the option to connect onboard electronics to a stationary data acquisition system. However, even if Jordan et al. [83] partially clarified that the coordination level trend is speed dependent, unfortunately the speed (m·s^−1^ or km·h^−1^) and physiological response (heart rate) were not reported and studied. Furthermore, in that study, the preferred walking speed was assessed on the treadmill only. The critical aspect concerning walking speed lies in the fact that each participant exhibits a distinct preferred walking speed. Therefore, the preferred walking speed should be assessed on overground to be more realistic [89], while the gait variability should be assessed on a treadmill [91] to overcome the GV setting. From this perspective, the aims of this study were as follows: (a) to assess individual preferred walking speed in overground setting; (b) to determine the physiological response and gait variability related to the different walking speeds on a treadmill.

## 2. Materials and Methods

### 2.1. Participants

Fourteen male students (age: 23.4 ± 2.7 years, body mass: 74.9 ± 6.8 kg, and body height 1.78 ± 0.05 m) voluntarily participated in this study. The participants were healthy, without any muscular, neurological, and tendinous injuries and did report that they were clear of any drug. The diet control in the pre-study was designed to eliminate the risk of any major differences between diets in total protein, carbohydrates, and saturated and unsaturated fats. After being informed on the procedures, methods, benefits, and possible risks related to study, all the participants provided their written informed consent. Experimental protocol was approved by the local ethics committee and was performed in accordance with the principles of the latest version of the Declaration of Helsinki.

### 2.2. Experimental Design

The testing sessions were conducted over four different days, separated by a five-day interval. Prior to the testing days, each participant underwent a 25 min familiarization period with the treadmill (two sessions). During the first two testing days (test–retest for the first test), participants walked outdoors three times at a self-selected speed (SS) on a linear flat asphalt surface for 1 km (average temperature 24.3 ± 1.2 °C and relative humidity of 18.2 ± 1.5%) between 10:00 and 12:00 a.m. An Apple Watch S. 7 (Apple^TM^, Cupertino, CA, USA) was worn to individually determine the SS in km·h^−1^ [92].

On the last two testing days, participants reported to a climate-controlled laboratory (23.5 ± 0.8 °C and 15.1 ± 1.3% for ambient temperature and relative humidity, respectively). In this session, they were asked to complete a 10 min warm-up and after, to walk on a calibrated treadmill (RAM 770 M, Arak, Iran) [90] at three different speeds: (i) equal to their SS determined in overground (SS); (ii) −20% of the SS (SS − 20); and (iii) +20% of the SS (SS + 20). Each speed condition was randomly administered and lasted 10 min, with 5 min of passive recovery in-between. Each participant was asked to wear the same running clothing and shoes (Cat. A3) in all of the testing sessions.

### 2.3. Measurement

During the walking test on a treadmill, heart rate (HR) was recorded continuously (Polar H-10, Kempele, Finland) and normalized as percentage of the maximal heart rate, %HR_max_ estimated [93] by (220—age). Participants also reported their rating of perceived exertion (RPE) on the CR10 scale immediately after completion of each walking speed. Kinematic data were obtained with an OptoGait system (sample rate—1000 Hz) and a specialized Software (Microgait^TM^, Bolzano, Italy) using a three-led filter (IN-OUT) [94]. Contact Time (CT), Swing Time (ST), Stride Length (SL), duration of Stride Cycle (SC), and GV, assessed as Phase Coordination Index (PCI), were determined.

The left–right coordination (phase coordination index, PCI) of walking gait was assessed according to Plotnik and coll. [13], normalizing the step time with respect to the stride time. The former relates to the time interval between a heel strike and the one of the contralateral leg, whereas the latter relates to the time interval between a heel strike and the consecutive one of the same leg. The normalization of step time with respect to the stride time determines the phase of the *i*-th stride (*ϕi*), which represents an index of bilateral coordination [13]. To preserve uniformity across all participants with possible different dominance, firstly, we calculated the average values of ST for both legs and used the leg with the higher ST as the reference for gait cycles. Successively, *ϕi* values for the other leg were computed as follows:
(1)$$\phi i=360{}^{\circ}\;\times \;\frac{t_{Si}-t_{Li}}{t_{L\left(i+1\right)}-t_{Li}}$$
where *t_Si_* and *t_Li_* denote the time of the *i*-th heel strike of the legs with the short and long ST, respectively, and *t_L(i+1)_* > *t_Si_* > *t_Si_*. The factors at the denominator of (1) relate to ST of the leg with the longest ST. Lastly, 360 was used to transform the variable into degrees [55]. A *ϕ* value of 180° indicates a successful walking symmetry, with step time being half of the gait cycle for each step. The GV encompasses the evaluation of the accuracy and consistency of phase generation and serves as the primary outcome. The accuracy level in phase generation, measuring how closely the series of generated phases align with the value 180°, was assessed by calculating the mean value of the absolute differences between the phase at each stride and 180°. This measure is denoted as *ϕ*_ABS:(2)$$\phi \_\mathrm{ABS}\;[{}^{\circ}]=\bar{\left|\phi_{i}-180{}^{\circ}\right|}.$$

To evaluate the level of consistency in phase generation across all strides for each participant, the coefficient of variation of the mean of *ϕ* was also determined. This consistency is represented as *ϕ*_CV [%]. Lastly, to compensate for the association between *ϕ*_ABS and *ϕ*_CV, the phase coordination index (PCI) was obtained as PCI = *ϕ*_CV + P*ϕ*_ABS, where P*ϕ*_ABS = 100 × (*ϕ*_ABS/180). Further details about the association between *ϕ*_ABS and *ϕ*_CV can be found in Plotnik and coll. [13]. Notably, the PCI provides insights into both the accuracy and consistency of phase generation.

### 2.4. Statistical Analysis

Results are expressed as mean ± standard deviation (SD). The Shapiro–Wilk test was used to verify the normality of the distribution. The reliability of the SS and PCI was assessed by an Intra-Class Correlation Coefficient (ICC) and classified as poor, if <0.05; moderate, if between 0.50 and 0.75; good, if between 0.75 and 0.9; and excellent, if >0.9, according to Koo and Li [95]. To assess differences for %HR_MAX_, RPE, CT, ST, SC, SL, PCI, *ϕ*_CV, *ϕ*_ABS, and *ϕ* over the 3 different walking conditions (SS − 20/SS/SS + 20%), a one-way, repeated-measures analysis of variance (RM-ANOVA) was used. When a significant F-value was found, post-hoc analysis (LSD) between conditions was performed. The ANOVA effect size was also calculated (partial eta squared $\eta_{p}^{2}$) and classified as small (<0.06); medium (0.06–0.14); and large (>0.14) [96]. The significance level was fixed as *p* ≤ 0.05. All of the analyses were conducted using the Statistical Package for Social Science software (V. 21.0, IBM SPSS Statistics, Chicago, IL, USA).

## 3. Results

The SS was 4.94 ± 0.58 (min/max: 4.00–6.20) km·h^−1^. The speed during SS − 20 was 3.96 ± 0.47 (min/max: 3.20–5.00) km·h^−1^ and the speed during SS + 20 was 5.80 ± 0.73 (min/max: 4.80–7.44) km·h^−1^. RM-ANOVA showed differences among the three walking speeds for RPE (0.84 ± 0.61, 0.97 ± 0.68, 1.78 ± 0.83 a.u. in SS − 20, SS, SS + 20%, respectively) as well as HR (50.95 ± 4.50, 52.85 ± 5.27, 56.70 ± 5.79%HR_max_, in SS − 20, SS, SS + 20%, respectively, with F_1,12_ = 24.680 and $\eta_{p}^{2}$ = 0.655 (ES: Large) with *p* = 0.005/F_1,12_ = 43.785 and $\eta_{p}^{2}$ = 0.814 (ES: large) with *p* = 0.035, respectively).

CT and SC decreased as the speed increased (Table 1 with post-hoc analysis) (F_1,12_ = 8.232 and $\eta_{p}^{2}$ = 0.388 (ES: Large) with *p* = 0.013/F_1,12_ = 4.974, and $\eta_{p}^{2}$ = 0.277 (ES: Large) with *p* = 0.044, respectively), while ST was unchanged (F_1,12_ = 0.898, and $\eta_{p}^{2}$ = 0.065 (ES: Medium) with *p* = 0.086). SL increased as the speed increased (F_1,12_ = 1146.447 and $\eta_{p}^{2}$ = 0.990 (ES: Large) with *p* = 0.0001). Conversely, the bilateral coordination (Table 2 with post-hoc analysis) data showed a parabolic trend. ANOVA showed large differences for *ϕ* (Figure 1), with F_1,12_ = 16.360 and $\eta_{p}^{2}$ = 0.561, with *p* = 0.001; PCI (Figure 2), with F_1,12_ = 17.731 and $\eta_{p}^{2}$ = 0.577, with *p* = 0.001; *ϕ*_CV, with F_1,12_ = 17.731 and $\eta_{p}^{2}$ = 0.833, with *p* = 0.001; *ϕ*_ABS, with F_1,12_ = 17.701 and $\eta_{p}^{2}$ = 0.577, with *p* = 0.001. The ICC for PCI was 0.905 (CI 95%: 0.704–0.969). The ICC for SS was 0.998 (CI 95%: 0.994–0.999).

## 4. Discussion

This study represents the first attempt to assess the individual preferred walking speed in 1 km overground, evaluating the physiological response and the gait variability on a treadmill. Our findings revealed a self-selected speed range between 4.00 and 6.20 km·h^−1^, which is a crucial finding with significant implications for future research on walking gait in healthy individuals. Previous studies have reported differences in physiological and perceptual responses when comparing treadmill walking to overground walking for self-selected speed attainment [24,25]. However, the practicality of outdoor walking is often hampered by various environmental obstacles, such as safety concerns and adverse weather conditions, prompting the use of treadmills in various physical activity programs as an alternative [93]. Nonetheless, the ecological validity of treadmill walking as a substitute for overground walking remains a relevant question that requires further investigation. Unfortunately, limited research has been dedicated to addressing this inquiry. Parvataneni et al. [26] discovered that treadmill walking at a self-selected pace demands a higher physiological response compared to overground walking, possibly due to increased co-contraction of agonist and antagonist muscles.

The second aim of this study was to assess the physiological response at different walking speeds. At the self-selected walking speed, the physiological response was 52.85 ± 5.27%HR_max_, which increased/decreased concurrently with the speed. In the current study, participants reported RPE ranging from 1 to 2, regardless of the environment setting, in accordance with previous laboratory-based studies that employed self-paced protocols [89,97,98]. Our investigation revealed significantly higher RPE during the SS + 20 on the treadmill session. According to Foster’s model of effort continua, this disparity in exertional perceptions [99] can be attributed to the moderate physiological demands associated with the walking speed.

The third aim was to determine the gait strategy variability at different speeds in the neighborhood of the overground self-selected speed. The kinematic data CT, SL, and SC showed a strong linear correlation when the speed increased (Table 1). Conversely, the motor coordination level showed a U-shaped behavior (Figure 2). Therefore, the gait variability was not conditioned by both fatigue effects and/or physiological efforts. Indeed, our investigation showed a U-shaped function (Figure 2) about the PCI as gait variability; thus, the PCI was higher (low coordination) at SS − 20 (3.81 ± 0.88%) and SS + 20 (3.36 ± 1.09%) compared to SS (2.64 ± 0.75%), with *p* = 0.001. The same parabolic U-shaped function trend (Table 2) was found for *ϕ*_CV, *ϕ*_ABS, and *ϕ,* according to Plotnik et al. [55].

A similar U-shaped trend in gait variability, analyzed using long-range correlation through detrended fluctuation analysis, has been observed in both running and walking gaits for female participants [14], although physiological effort measurements were not included. In both walking and running conditions, Jordan et al. [14,29] found that the lowest gait variability was observed at 100–110% of the preferred walking speed and 100% of the running speed, respectively. Therefore, the PCI and the long-range correlation using detrended fluctuation analysis might be equivalent in assessing gait variability from a methodological standpoint. However, the PCI analysis provides more comprehensive information regarding gait variability, including the accuracy and consistency of phase generation (*ϕ*_CV, *ϕ*_ABS, and *ϕ*). These metrics quantify the ability of young males to coordinate left–right stepping on flat terrain at different speeds and evaluate the precision and coherence of the gait pattern [13].

These metrics was able to quantify the ability of young males to coordinate left–right stepping on flat terrain at different speeds. Simultaneously, the PCI assesses both the precision (Table 2) of anti-phase coordination and the coherence of the gait pattern [55]. Considering that the lower limb during the walking gait is not constrained, as in pedaling on a bicycle [1], we think that it is more appropriate to consider all kinematic parameters (stride cycle, stride length, swing time, and contact time) as in PCI, compared to long-range correlation using detrended fluctuation analysis where only the stride cycle is used.

## 5. Conclusions

Our study demonstrated that walking at a specific self-selected pace requires the participant to continuously adjust the force produced and its timing relative to the foot position [1]. Reasonably, when the timing or the module of the force is not applied appropriately, an unwanted acceleration or deceleration of the lower limb occurs, inducing a fluctuation in cycle duration. It is possible that unusual riding positions change cycling variability due to mechanical factors [83]. Therefore, an increase in the number of corrections of the leg/foot velocity through timing activation of lower leg muscles is expected to increase gait variability, possibly as a function of walking speed. The gait variability is believed to reflect the need for central pattern generators to correct the timing activation of different muscles throughout the step cycle. As such, it is possible that the increase in the variability observed in SS − 20 and SS + 20 reflects a higher number of corrections during the cycle due to the position [88,100]. This is also suggested by Marck et al. [101], who observed that restricting arm movements altered hip movement variability during walking. In conclusion, the physiological response and kinematics variables changed linearly when the speed increased. The walking gait coordination followed U-shaped curves as a function of walking speed; the better gait coordination was at self-selected speed in healthy young males. These findings support the hypothesis that reducing PCI at SS is reflective of enhanced stability of these speeds. Therefore, SS could be a new methodological approach to choose the individual walking speed in overground, to normalize the intensity of the speed, and to avoid a gait pattern alteration. For future perspectives, we aim to expand this protocol to encompass diverse surfaces and a wide range of age groups (including both young and elderly individuals), as well as individuals with injuries and lower limb conditions.

## Figures and Tables

**Figure 1 jfmk-08-00158-f001:**
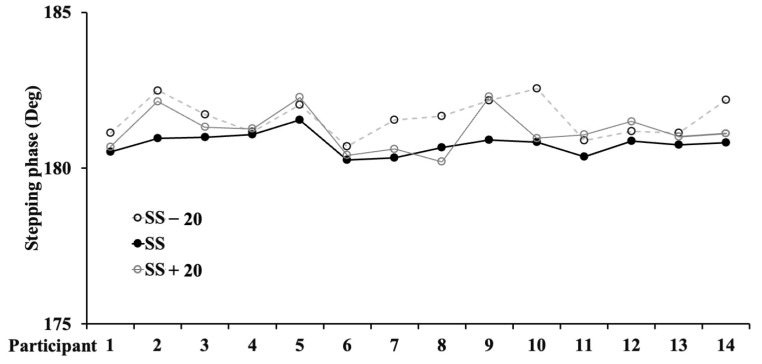
Stepping phase value for each participant on the three different speeds.

**Figure 2 jfmk-08-00158-f002:**
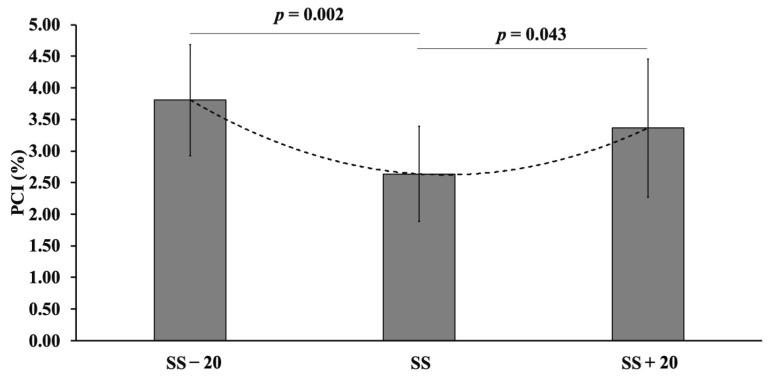
Bar chart demonstrating the PCI differences between the three different speeds.

**Table 1 jfmk-08-00158-t001:** Footstep variables on the three different walking speeds (SS − 20/SS/SS + 20).

Variables	SS − 20	SS	SS + 20
CT (s)	0.80 ± 0.07 ^†,‡^	0.68 ± 0.06 ^$^	0.59 ± 0.06
ST (s)	0.40 ± 0.03	0.39 ± 0.03	0.40 ± 0.03
SL (cm)	131 ± 11 ^†,‡^	147 ± 10 ^$^	163 ± 11
SC (s)	1.14 ± 0.21 ^‡^	1.08 ± 0.07	0.99 ± 0.07

Contact Time (CT); Swing Time (ST); Stride Length (SL); Stride Cycle (SC) are reported as mean and standard deviation (SD). Significant (*p* < 0.05) differences between SS − 20/SS are denoted as “†”, SS − 20/SS + 20 as “‡”, SS/SS + 20 as “$”.

**Table 2 jfmk-08-00158-t002:** Effects of the three different walking speeds (SS − 20/SS/SS + 20) on the bilateral coordination parameters.

Variables	SS − 20	SS	SS + 20
PCI (%)	3.81 ± 0.88 ^†^	2.64 ± 0.75 ^$^	3.36 ± 1.09
*ϕ*_CV (%)	2.04 ± 0.47 ^†^	1.45 ± 1.83 ^$^	1.83 ± 0.58
*ϕ*_ABS (deg)	1.77 ± 0.42 ^†^	1.19 ± 0.33 ^$^	1.55 ± 0.51
*ϕ* (deg)	182 ± 0.6 ^†^	181 ± 0.3 ^$^	181 ± 0.7

Note: data are expressed as mean and SD; significant (*p* < 0.05) differences between SS − 20/SS are denoted as “†”, SS/SS + 20 as “$”.

## Data Availability

The data that support the findings of this study are available from the corresponding author upon reasonable request.

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
