# Peer review of "Gait Variability at Different Walking Speeds"

_jfmk, 2023, doi:10.3390/jfmk8040158_

Round 1
Reviewer 1 Report
Comments and Suggestions for Authors
I am unsure if I understand the research questions that are addressed in this manuscript. As the authors’ motivation is to determine metabolic demands and gait variability. Yet, no data is presented with metabolic changes? Do the authors claim that PCI is related to metabolic expenditure? In case yes, please explain it.
If the question regards the preferred walking speed on the treadmill and overground, it has already been addressed by previous publications, especially the referred paper by Dasilva et al. 2010. What is the message of this paper that is different than previous publication? And what is it adding to the knowledge we already have?
Is formula 1, part of the measurements, or is it calculated values? It might be a type as tl(i+1)>tsi>tli. If 180 degrees is a symmetry between step time, it means that tsi+tli=2tL(i+1). Does it mean that tsi=tli? Is that what the authors call as “successful walking symmetry”?
There are also some data gathering, like HR and RPE that are just mentioned as numbers, but it could be interesting to show these in relation to PCI. Did the authors measure other physiological parameters like blood pressure (especially systolic pressure), or core temperatures that could also be related to metabolic expenditure?
If the subjects are wearing shoewear cat.type 3, it means a high cushioning factor is involved. How does this influence the ST and “Øi”?
Phase in most scientific publications is not referred to as Ø, but rather Phi in Latin ( ϕ). Please use the same mathematical term to avoid confusion.
Comments on the Quality of English LanguageThere is a need for proofreading of the paper. Line 55, "anyway" should prbably be however. further it was shown....etc.
Author Response
Reviewer 1
I am unsure if I understand the research questions that are addressed in this manuscript. As the authors’ motivation is to determine metabolic demands and gait variability. Yet, no data is presented with metabolic changes? Do the authors claim that PCI is related to metabolic expenditure? In case yes, please explain it. If the question regards the preferred walking speed on the treadmill and overground, it has already been addressed by previous publications, especially the referred paper by Dasilva et al. 2010. What is the message of this paper that is different than previous publication? And what is it adding to the knowledge we already have?
R: Thanks for this comment. The paper did not present any data regarding metabolic changes. We reported heart rate as a surrogated parameter for metabolic demand, considering the linear relationship between heart rate and oxygen consumption. Based on your suggestion, we have replaced "metabolic demand" with "physiological response?”
In contrast to Da Silva et al. (2020) who investigated the differential effects of environmental settings (treadmill vs overground) on physiological, perceptual, and affective responses to exercise at a self-selected pace, our aim (Line 75) was to evaluate the heart rate and the gait variability in relation to the different walking speeds on a treadmill, after assessing individual preferred walking speed in an overground setting. and b).
In light of this, the novelty of this paper (as indicated in Line 28 and 258-263) is that a self-selected walking speed could represent a new methodological approach for determining individual walking speeds, normalizing speed intensity, and preventing alterations in gait patterns.
Is formula 1, part of the measurements, or is it calculated values? It might be a type as tl(i+1)>tsi>tli. If 180 degrees is a symmetry between step time, it means that tsi+tli=2tL(i+1). Does it mean that tsi=tli? Is that what the authors call as “successful walking symmetry”?
R: Thanks for this question. The formula 1 is calculated. According to the Plotnik et al (2007), the “successful walking symmetry” refers to achieving step times that are exactly half of the gait cycle for each step, and it is associated with the ϕ value. Hence, it is correct to state that if 180 degrees is a symmetry between step time, it means that tsi+tli=2tL(i+1) and tsi=tli.
.
There are also some data gathering, like HR and RPE that are just mentioned as numbers, but it could be interesting to show these in relation to PCI.
R: Thank you for this suggestion. The relationships between physiological effort (HR-RPE)) and PCI is missing because RPE and HR ncreased at the increased the speed (Line 21 (updated), Line 159-160, Line 206-207) while PCI showed a U-shaped function (Figure 2), thus: PCI was at self-selected speed (SS) and higher at lower/higher speeds compared to the SS (Line 24-25, Table 2, Line 176-177, Line 219-222)
Did the authors measure other physiological parameters like blood pressure (especially systolic pressure), or core temperatures that could also be related to metabolic expenditure?
R: Thanks for this question. We did not measure other physiological parameters. However, it could be useful for future research to add other parameters, like blood pressure or core temperatures.
If the subjects are wearing shoewear cat.type 3, it means a high cushioning factor is involved. How does this influence the ST and “Øi”?
R: Thanks for this question. We intend to use the same types of shoes to enhance the replication of this study. Nevertheless, the high cushioning factor, which promotes running (but not walking), is attributed to certain intrinsic characteristics, such as increased stiffness in the shoe sole
Phase in most scientific publications is not referred to as Ø, but rather Phi in Latin (ϕ). Please use the same mathematical term to avoid confusion.
R: Thanks for this suggestion, We have replaced Ø with “ϕ”.
Reviewer 2 Report
Comments and Suggestions for Authors
Author reported the assessment of gait variability of an individual based on their walking speed and attempt to determine their relation with their metabolic demand. I believe this manuscript presented the good way to determine an individual way of walking at their own speed. This may or may not help reducing the risk of injury based on the roughness of terrain. However, I found that study is missing few different aspect to fully understand the gait variability.
1. Author designed the experiment on linear asphalt surface and treadmill. It would be better if author can include the relationship between surface roughness and gait variability. How can this study be applied to rough surface to fully understand the risk of injury related to the way an individual walks?
2. If there is a sport related injury, how can we use this study to help understand the recovery roadmap of an individual.
3. Please discuss how to study gait variability of different gender and people with age group of late-20s or early to mid-30s. Also, it would be better if author discuss if we can use this study on elderly or people with injury.
Author Response
Reviewer 2
Author reported the assessment of gait variability of an individual based on their walking speed and attempt to determine their relation with their metabolic demand. I believe this manuscript presented the good way to determine an individual way of walking at their own speed. This may or may not help reducing the risk of injury based on the roughness of terrain. However, I found that study is missing few different aspect to fully understand the gait variability.
Author designed the experiment on linear asphalt surface and treadmill. It would be better if author can include the relationship between surface roughness and gait variability. How can this study be applied to rough surface to fully understand the risk of injury related to the way an individual walks?
R: Thanks for this comment. In the future, we intend to investigate gait variability associated with various surfaces, such as sand, track and field, asphalt, grass, and pavement. This study represents the initial step in a roadmap for examining walking gait on different surfaces, within diverse age groups, and across varying gradients.
If there is a sport related injury, how can we use this study to help understand the recovery roadmap of an individual.
R: Thanks for this comment. Given the established evidence that individuals with lower limb injuries exhibit increased variability (Armitano-Lago et al., 2023), this study could prove beneficial in the recovery roadmap by aiding in the individual selection and adjustment of walking speeds to reduce PCI and enhance stability.
Please discuss how to study gait variability of different gender and people with age group of late-20s or early to mid-30s. Also, it would be better if author discuss if we can use this study on elderly or people with injury.
R: Thanks for this comment. As future perspectives, we aim to expand this protocol to encompass diverse surfaces, a wide range of age groups (including both young and elderly individuals), as well as individuals with injuries and lower limb conditions. (lines 263-265)
Round 2
Reviewer 2 Report
Comments and Suggestions for Authors
Author's response does not clarify the issues raised. It is strongly recommended to read the comments and respond accordingly.
Despite reporting different quantification methods to observe the change in gait pattern, I believe the design of experiment lacks studying all the variables to conclude the study. The study was conducted on healthy male students and was concluded that self-selected speed is new methodology to avoid altered gait pattern, whereas self selected speed is individual's own choice. If author does not want to study on different terrains or conditions then I believe it would be best for the interest of readers to include people with injury and elderly people in the study. Also, please include physiological parameters other than HR to demonstrate the metabolic demand such as respiration rate, SpO2, blood pressure etc. to help understand better. What is the range of self selected speed?
Author Response
Author's response does not clarify the issues raised. It is strongly recommended to read the comments and respond accordingly.
R: dear Reviewer thanks for this suggestions, we provide for each point several clarifications, Anyway Your support was partially useful to improve our manuscript
Despite reporting different quantification methods to observe the change in gait pattern, I believe the design of experiment lacks studying all the variables to conclude the study.
R: In this study We used one method only, to observe the change in gait pattern; thus, the phase coordination index (Line 105-141). The experimental design was performed to clarify if the walking gait variability is related to the walking speed. As demonstrated by the results at self-selected speed there is a better bilateral coordination.
The study was conducted on healthy male students and was concluded that self-selected speed is new methodology to avoid altered gait pattern, whereas self selected speed is individual's own choice.
R: in our case is not a new method but a method to avoid altered gait pattern based on self-selected speed. In fact, in line 28, 60, 261 was reported “methodological approach”
If author does not want to study on different terrains or conditions, then I believe it would be best for the interest of readers to include people with injury and elderly people in the study.
R: the aims (Line 75-77) of this study were: a) to assess individual preferred walking speed in overground setting, b) to determine the metabolic demand and gait variability related to the different walking speeds on a treadmill. For the future could be interesting to study people with injury and elderly people, but we cannot overcome this point because the aim of our Laboratory is to investigate young healthy people and sport performance, only.
Also, please include physiological parameters other than HR to demonstrate the metabolic demand such as respiration rate, SpO2, blood pressure etc. to help understand better.
R: the scientific evidence showed that:
- Respiration rate is an important indicator of a person's health, and thus it is monitored when performing clinical evaluations but not to assess “Metabolic demand”
Al-Khalidi FQ, Saatchi R, Burke D, Elphick H, Tan S. Respiration rate monitoring methods: a review. Pediatr Pulmonol. 2011 Jun;46(6):523-9. doi: 10.1002/ppul.21416
- Oxygen saturation (SpO2) is the percentage of hemoglobin that is saturated with oxygen, converting it to oxyhemoglobin, anyway the literature showed that the SpO2
could be used indirectly to assess the high exercise intensity (Batterson 2023) not for metabolic demand; anyway, in our study the exercise intensity was lower (Line 160, 50-55 %HRMAX)
Batterson PM, Kirby BS, Hasselmann G, Feldmann A. Muscle oxygen saturation rates coincide with lactate-based exercise thresholds. Eur J Appl Physiol. 2023 Oct;123(10):2249-2258. doi: 10.1007/s00421-023-05238-9
- Blood pressure (BP) is a mandatory safety measure during graded intensity clinical exercise stress testing (Sharman 2015). In our study the exercise intensity was lower (Line 160, 50-55 %HRMAX)
Sharman JE, LaGerche A. Exercise blood pressure: clinical relevance and correct measurement. J Hum Hypertens. 2015 Jun;29(6):351-8. doi: 10.1038/jhh.2014.84
We are very happy if the reviewer can provide scientific evidence about the respiration rate, SpO2, blood pressure, are able to assess “Metabolic demand” more than heart rate in walking gait at lower intensities.
Anyway, the literature is consistence about the use of heart rate to assess metabolic demand/ physiological response on walking gait:
Karvonen J, Vuorimaa T. Heart rate and exercise intensity during sports activities. Practical application. Sports Med. 1988 May;5(5):303-11. doi: 10.2165/00007256-198805050-0000
To avoid some misleading interpretation with changed metabolic demand with physiological response
What is the range of self selected speed?
R: the range of self-selected speed was included by the first version in September 9, 2023 (Results section – Line 156; The SS was 4.94 ± 0.58 (min/max: 4.00 - 6.20) km·h-1)
Round 3
Reviewer 2 Report
Comments and Suggestions for Authors
Authors responded to the comments raised clearly now, I am satisfied with the responses. The manuscript can be accepted for publication in present form.